# Quantifying participant distress: Validity and applicability of a distress measure to evaluate harm in quantitative assessments

Jess MacArthur[1]*, Ratan Budhathoki[2], Min Prasad Basnet[2], Ambika Yadav[2], Sabitra Dhakal[2], Juliet Willetts[1]

**1** Institute for Sustainable Futures, University of Technology Sydney, Sydney, New South Wales, Australia, **2** SNV Nepal, Kathmandu, Nepal

* jessica.macarthur@uts.edu.au

## Abstract

Structured interviews remain a key approach to collect information from community members, particularly in development contexts. Such enumerated surveys often focus on potentially distressing topics including gender equality, social inclusion, well-being, and even socio-demographics. Researchers have an obligation to consider the ethics of survey processes and mitigate potential distress for participants. However, approaches to quantify and evaluate participant distress remain nascent outside of clinical practice. To support ethical considerations in quantitative survey deployment, we introduce a four-item formative measure to analyze interview ease, stress, privacy, and comprehension. We present the measure's conceptual and empirical development and examine the validity of the measure through data from Cambodia and Nepal (n = 4,674) using Partial Least Squares Structural Equation Modeling (PLS-SEM) for formative measurement model assessment. The measure is shown to have content and face validity, anticipated divergence with two reflective constructs, low collinearity, structural validity, and construct validity through known groups testing. As ethical considerations are increasingly recognized as important in research in both development and other research and evaluation contexts, tools to diagnose and analyze distress can support in mitigating negative impacts.

## Introduction

Ethical considerations in survey design continue to be explored and scrutinized in evaluation and research. Key considerations include privacy and confidentiality, utilizing clear questions, ensuring informed consent, and avoiding persuasion or pressure [1]. Ultimately, these considerations aim to uphold ethical principles and mitigate potential distress [1,2].

Within international development activities, enumerated surveys remain a popular way of collecting data at scale [3,4] and focus on answering research questions such

**Data availability statement:** All relevant data are within the paper and its Supporting Information files.

**Funding:** This work was supported by the Australian Government's Water for Women Fund [Grant number WRA-034 - JW]. The funders did not play any role in the study.

**Competing interests:** The authors have declared that no competing interests exist.

as "how many" and "to what extent". They are primarily conducted using structured interviews in which enumerators mark down responses either on paper on using digital tools and surveys can take up to several hours to conduct. Questions are often multiple-choice which allows the survey to produce quantitative results by converting the multiple-choice responses to numeric values. Sample sizes are often larger than qualitative studies to ensure statistical power and generalizability of results [5].

Ethical considerations are even more important in surveys which focus on potentially sensitive topics including health, equality, inclusion, gender, wellbeing and socio-demographics. Discussing these sensitive topics can lead to reactions including acute stress, anxiety, depression, or embarrassment [6]. In tangible terms participants can become emotional, become stressed, experience a change in mood, feel uncomfortable, and be unable to finish the survey – these responses are called participant "distress" [1,2].

In the United States and Australia, distress assessments have been used in psychological research related to traumatic experiences [7–9], research coercion within vulnerable populations [9], research related to post-traumatic stress syndrome [10], and clinical studies with adults and children [11–13]. These existing tools (summarized in S1 Table) range from 1 to 34 items with varying levels of measure validity testing. However, these existing tools: 1) have been designed for use in hospital settings for psychiatric, psychological and clinical research contexts, with less applicability in sociological studies; 2) rely on self-facilitated questionnaires in hospital settings, rather than in enumerator household-based surveys in rural communities; and 3) have not been designed for or tested in international development contexts, including south and southeast Asia.

As such, this paper presents the validation of a rapidly designed distress measure to track and evaluate distress in enumerated quantitative surveys within sociological research in international development contexts. The article continues with a brief description of the methods used to develop and validate the distress measure as part of three survey deployments in Cambodia and Nepal (n = 4,674). The results of the validation analysis follow. We then discuss three potential use cases of the measure and reflect on its limitations. We conclude by describing opportunities to continue refining and using the measure in enumerated surveys.

## Methods

The design and validation of the distress measure emerged in the context of the Water, Sanitation and Hygiene – Gender Equality Measure (WASH-GEM) which measures changing gender norms, dynamics, and structures within WASH programming [14–16]. The development of the full WASH-GEM tool followed robust instrument design processes and validation procedures for social measures through rapid, exploratory and validation pilots [16,17]. As the WASH-GEM focuses on sensitive topics, it requires a high ethical standard and robust tracking of potential distress. Validating the tool's distress items was a secondary outcome of the WASH-GEM's development.

The catalyst for including and testing distress-related items came during the rapid pilot phase of the WASH-GEM's development [16]. During the WASH-GEM's cognitive interviewing process in Nepal, the researchers noted that participants – both men and women – had emotional responses to survey items not expected to necessarily cause distress, including questions related to self-efficacy, equality awareness, control over resources, and physical wellbeing. During a daily debrief, the team decided it would be important to systematically track participant distress for all survey respondents with the WASH-GEM going forward and a module was designed overnight for the next day's pilot deployment.

As the distress measure was developed for immediate deployment to track distress during the pilot activity, it necessitated an abridged design process. Additionally, as the measure was added to an existing survey instrument, it was not feasible to test a long list of items, rather to identify a short list of items for testing.

The designed distress measure is formative rather than reflective as each of the four items represents one aspect of distress and is documented by enumerator observations rather than respondent responses or self-assessment, in alignment with best practice checklists [18,19]. As such, the traditional scale (reflective measure) validation tools are not relevant to this process, instead leading to validation procedures for formative measures [20,21].

## Item identification and refinement

The four items for testing were identified during a team debrief workshop in the rapid pilot phase of the WASH-GEM. The items were developed to align with the WASH-GEM's distress protocol built on academic best practices for research on sensitive topics [22]. Through brainstorming discussions and review of the study's distress protocol the team clarified that participant distress was caused by four key factors: the smoothness of the interview process (ease), the stress of the participant often shown through emotional reactions (stress), the privacy of the interview (privacy), and the comprehension of the participant related to the complex topics covered in the rest of the survey (comprehension). The six team members who participated in this brainstorming process included a mixed gender group of expert academics and practitioners with expertise in monitoring, evaluation, do-no-harm approaches, gender equality and social inclusion, and instrument design. The structure of the items was then drafted to mirror existing items within the wider WASH-GEM tool which was in the process of cognitive testing.

Each of the four items aimed to engage with a different domain of do-no-harm often explored within cognitive interviewing and distress protocol development. *Ease* explores the extent to which the interview questions flow and is related to the rapport between the respondent and interviewer [23]. *Stress* reflects the experiences of adverse emotional reactions that might arise during the interview [22]. *Privacy* relates to the experience of participants in being able to share their answers freely and openly without the influence of others – including family members [24,25]. Lastly, *comprehension* reflects the importance of interviewees understanding the questions and quantifies the requirement of enumerators to explain items [23].

The tested items are summarized in Table 1 with the notes provided to enumerators to clarify response options. The team additionally included a text box for any further open-response reflections from the enumerators. Responses from this item (ob5) are not included in this analysis and are a topic for future exploration. Further details on the questionnaire and WASH-GEM can be found on the WASH-GEM Learning Website (https://waterforwomen.uts.edu.au/wash-gem/).

## Content and face validity

Between the rapid and validation pilots of the wider WASH-GEM tool, the distress measure's items were reviewed in the context of relevant literature, reviewed by experts and discussed with enumerators. After the rapid pilot, the team conducted a literature review to ensure alignment with best practice in social research which led to the refinement of the items. The items were also reviewed by a range of global and local experts (n = 22) on survey design, do-no-harm, inclusion and ethical research processes. Additionally, the team conducted three workshop discussions with enumerators

**Table 1. Distress measure item summary.**

| Item Number | Item | Responses and Enumeration Notes | Conceptual Purpose |
|---|---|---|---|
| Ob1. Ease | How easy was this interview for you to conduct? | • Very easy – No tension or challenges in connecting<br>• Easy – A little tension<br>• Difficult – Some tension<br>• Very difficult – A lot of tension | Quantifies the general ease between the respondent and enumerator including rapport and tension. |
| Ob2. Stress | What was the stress level of the participant? | • None – No visible signs of stress<br>• Minimal – Short or passing signs of stress<br>• Mild – Mild stress having to pause the interview once<br>• Severe – Significant stress having to pause the interview multiple times | Identifies the level of stress for the respondent during the interview as viewed through visible emotional responses. |
| Ob3. Privacy | Was the context private? | • Very much – Fully private<br>• Somewhat – Some visitors who didn't stay<br>• Not really – Visitors who stay, but remain quiet<br>• Not at all – Visitors who interrupt and engage | Clarifies the level of privacy of the interview (do-no-harm and encouraging honest answers). |
| Ob4. Comprehension | Did the participant understand the questions? | • Very much – No clarity required<br>• Somewhat – A little clarity provided<br>• Not much – Some clarity provided<br>• Not at all – A lot of clarity provided | Describes the extent to which questions had to be explained and clarified to the respondent. |
| Ob5. Enumerator Reflections | Please provide any other comments or observations. | [Text box] | Provides opportunity for further qualitative reflections. |

(n = 40) who had enumerated many surveys with respondents in their respective contexts. These three steps helped to refine the items, but did not led to any significant changes or additions to the initial four items, only adjustments to the wording of the response options.

## Validation datasets and sampling

The distress measure was validated using three cross-sectional data sets (n = 4,674) from the WASH-GEM in Cambodia and Nepal. The first two datasets were from a baseline project assessment in two Nepali districts (Sarlahi and Dailekh) and three Cambodian provinces (Kampong Thom, Prey Veng, and Kandal). The last dataset was from a midline assessment in Nepal also from Sarlahi and Dailekh. In each dataset, the survey tool aimed to interview a dyad of the male primary decision-maker (amongst men) and the female primary decision-maker (amongst women), as such the datasets included a near equal number of male and female respondents. The survey was conducted in Nepali and Khmer in Nepal and Cambodia respectively, through cross-checked translations of the tool. In-depth enumerator training was conducted in accordance with the WASH-GEM ethical guidelines including distress protocols and agreement on specific translations with regards to local dialects. The deidentified data can be found in S1 Data.

All three datasets relied on stratified multistage random sample design, ensuring a balanced sample of women and men respondents. Focused on the five target districts/provinces, the team then leveraged the purposeful selection of sub-districts/municipalities to represent the breadth of variance in program catchment areas, using Demographic and Health Survey data to identify variations in WASH status, electricity, land ownership, poverty levels, and female-headed households. Next, the team randomly selected communes/wards from lists of program working areas; lastly, the team randomly selected villages. In each commune/ward, two villages were selected: one primary and one alternate if achieving just over 80 respondents per village was not feasible. At the village level, enumerators sought socio-economic variation in their selection of households. In Nepal this process was done twice, with villages randomly selected for the two different phases of data collection; however the selected wards remained the same.

Data was pooled from the three datasets in Cambodia and Nepal and missing data was replaced using imputed means by gender and region. As only a small number of respondents identified their gender as other, a subset of male-female data was used for gender comparisons to avoid drawing unsubstantiated conclusions.

## Measure validation

To validate the *Distress-4* measure for more widescale use, we utilized Partial Least Squares Structural Equation Modeling (PLS-SEM) and bootstrapping (5,000 iterations). Procedures broadly aligned with work by Hair et al. [26]. In this approach, a formative measure is validated by creating a structural equation model with other strategically selected variables from the WASH-GEM, allowing the model to be identified. Literature describes three ways to include additional variables to identify the model and validate a formative measure: 1) one or two reflective indicators that summarize the same concept as the formative measure; 2) two reflective constructs theoretically related to the formative measure; or 3) a combination of one global indicator and one construct [20,26]. Our datasets did not include a global indicator or alternative reflective indicators of distress, but the datasets did include a variety of other constructs within the distress measure's nomological network; as such we have selected option two for model identification. PLS-SEM was selected as a modeling approach, as it is appropriate for a mixture of formative and reflective measures and can utilize non-normally distributed data [26]. Data analysis was conducted in RStudio including the use of the SEMinR package [26,27].

*Reflective Variable Identification.* The model was identified using two theoretically and statistically related reflective constructs [20]. The wider WASH-GEM tool comprises 17 validated measures including a range of both formative and reflective constructs [16]. For six of the WASH-GEM's most valid reflective constructs, there is a theoretical causal relationship with distress; lower scores in these themes theoretically lead to increased distress (*Household Influence, Household Autonomy, Self-efficacy, Collective Action, Equality Awareness* and *Mobility*). This pattern was also observed during the WASH-GEM's rapid pilot, in that individuals with lower scores in these measures were observed to have increased distress. To select two reflective constructs to use for model identification, intra-measure Pearson correlation coefficients were calculated and plotted on a correlation matrix for the six relevant measures and a simple sum version of the distress measure. From these six measures, two were selected for the model. In a slight variation from existing practice, instead of using the reflective constructs as outcome variables, the directionality was reversed to better align with theoretical and observed insights. The results of this reversed model were cross-checked against the initial model to ensure this choice did not impact the validation results.

*Divergent Validity.* The two reflective constructs used to identify the PLS-SEM model were expected to have weak-to-moderate divergence from distress measure scores. As part of the model, path coefficients were calculated for each of the two connections, estimating that there would be weak-to-moderate negative path coefficients ($\beta \leq -0.20$, $p \leq 0.01$) between the reflective constructs and the formative construct [27].

*Item Collinearity.* For the measure to effectively operate formatively, there should not be a high level of collinearity between the items. Item collinearity was first assessed through intra-item correlations and plotted on a correlation matrix. Item collinearity was next assessed as part of the PLS-SEM model through variance inflation factors (VIF). A range of 0.2 to 0.6 was adopted as the thresholds for the intra-item correlations, and a threshold of 3 was adopted for the VIF values [26] to indicate low collinearity – required for a formative measure.

*Item Significance and Relevance.* Item significance and relevance were assessed as part of the PLS-SEM model for each of the distress measure's four items. The statistical significance of each item's weight was assessed ($p \leq 0.05$) alongside loadings, with loadings $\geq 0.50$ justifying the inclusion of the item [26]. For items with statistically insignificant weights and loadings $\leq 0.50$, inclusion could be justified through statistical significance of the loading and theoretical relevance [26].

*Structural Model.* Although the main output of the analysis was to assess the validity of the distress measure, the model also tested the influence of the two reflective constructs (*Equality Awareness* and *Self-efficacy*) on respondent

distress for the case study dataset. From theory and observation, it was anticipated that the overall model would have a weak coefficient of determination ($0.20 < r^2 < 0.30$), for the study dataset. In alignment with PLS-SEM best practices, model fit statistics are not required, instead relying on the path coefficients and the coefficient of determination [26].

*Scoring*. A comparison was done to identify which scoring approach would be most appropriate – a simple sum or weighted sum. While weighted sums from regressions are more accurate, there is a strong benefit to project teams in being able to quickly score responses, without the use of statistical analysis software. As such, we assessed the correlation between the weighted sum through the PLS-SEM model and a simple sum scoring. A Pearson correlation coefficient greater than 0.90 with $p \leq 0.01$ was deemed to provide sufficient rationale for a simple sum score. This was also tested for each of the three datasets by gender.

*Construct Validity*. Construct validity explored the extent to which the measure demonstrated the expected theoretical empirical relations, using the simple sum version of the distress measure. Using known groups t-testing, we tested the hypothesis that poorer, older, and less educated people would have higher levels of distress than their wealthier, younger and higher educated counterparts. These groups were identified through observations of the types of people who experienced distress during the rapid pilot and discussions with the program managers and enumerators.

While not included in this article due to ethical constraints, the same validation procedures were also conducted on a larger sample (total n = 6,025) with additional responses from Bhutan, Laos, and Ghana. The same overall validation results were found in this larger sample.

## Ethical considerations

The study was approved in two phases by the University of Technology Sydney (UTS HREC REF NO. ETH18–2599 – Project 17232 and 21051). The first phase covered datasets 1 and 2, and the second phase covered dataset 3. The studies aligned with the WASH-GEM do-no-harm processes and distress protocols. Informed consent was obtained verbally by each participant prior to the start of each interview, as indicated in the approved consent procedures and recorded digitally through a checkbox checked by the enumerator. Additional information regarding the ethical, cultural, and scientific considerations specific to inclusivity in global research is included in the Supporting Information (S1 Checklist).

## Results

### Sociodemographic characteristics

Sociodemographic characteristics of the three datasets are found in Table 2. The participants came from five different provinces/districts of Cambodia and Nepal and were a near even split between male and female respondents. Overall, nearly half of the responses had preschool or less level education and were between 31 and 49 years of age. Wealth quintiles were calculated using principal component analyses within each country and dataset using a series of asset-based questions. There was minimal missing data for the socio-demographic responses.

### Distress measure items

The four identified items for the distress measure relate to interview ease (ob1), stress (ob2), privacy (ob3), and comprehension (ob4) as introduced in Table 3. Overall, the majority of enumerators identified that interviews were easy or very easy to conduct, that the participants did not display physical signs of stress, that the interviews were private and that comprehension was quite high. However, all four items did have responses that indicated distress of participants.

### Intra-item and measure collinearity

Intra-measure and intra-item correlation coefficients were calculated and plotted on a correlation matrix (Fig 1), with two objectives: 1) to identify potentially appropriate WASH-GEM measures to use within the PLS-SEM model and 2) to

**Table 2. Sociodemographic characteristics of the study participants in Cambodia and Nepal.**

| | Cambodia Baseline (N = 1496) | Nepal Baseline (N = 1558) | Nepal Midline (N = 1620) | Overall (N = 4674) |
|---|---|---|---|---|
| **Province/District** | | | | |
| Kampong Thom Province | 502 (33.6%) | 0 (0%) | 0 (0%) | 502 (10.7%) |
| Kandal Province | 498 (33.3%) | 0 (0%) | 0 (0%) | 498 (10.7%) |
| Prey Veng Province | 496 (33.2%) | 0 (0%) | 0 (0%) | 496 (10.6%) |
| Dailekh District | 0 (0%) | 626 (40.2%) | 632 (39.0%) | 1258 (26.9%) |
| Sarlahi District | 0 (0%) | 932 (59.8%) | 988 (61.0%) | 1920 (41.1%) |
| **Gender** | | | | |
| Women | 748 (50.0%) | 801 (51.4%) | 808 (49.9%) | 2357 (50.4%) |
| Men | 748 (50.0%) | 757 (48.6%) | 812 (50.1%) | 2317 (49.6%) |
| Other/Missing | 0 (0%) | 0 (0%) | 2 (0.1%) | 2 (0.0%) |
| **Wealth** | | | | |
| Poorest | 435 (29.1%) | 379 (24.3%) | 421 (25.9%) | 1235 (26.4%) |
| Poor | 380 (25.4%) | 483 (31.0%) | 249 (15.3%) | 1112 (23.7%) |
| Mid | 364 (24.3%) | 366 (23.5%) | 344 (21.2%) | 1074 (22.9%) |
| Rich | 204 (13.6%) | 169 (10.8%) | 296 (18.2%) | 669 (14.3%) |
| Richest | 113 (7.5%) | 161 (10.3%) | 316 (19.4%) | 590 (12.6%) |
| Missing | 1 (0.1%) | 2 (0.1%) | 0 (0%) | 3 (0.1%) |
| **Education Groups** | | | | |
| Preschool- | 263 (17.6%) | 1001 (64.2%) | 1046 (64.6%) | 2310 (49.4%) |
| Primary | 810 (54.1%) | 273 (17.5%) | 315 (19.4%) | 1398 (29.9%) |
| Secondary+ | 423 (28.3%) | 284 (18.2%) | 259 (16.0%) | 966 (20.7%) |
| **Age Groups** | | | | |
| 18-30 | 166 (11.1%) | 266 (17.1%) | 177 (10.9%) | 609 (13.0%) |
| 31-49 | 685 (45.8%) | 764 (49.0%) | 818 (50.2%) | 2267 (48.4%) |
| 50+ | 646 (43.2%) | 530 (34.0%) | 631 (38.8%) | 1807 (38.6%) |
| Missing | 0 (0%) | 0 (0%) | 2 (0.1%) | 2 (0.0%) |

explore intra-item correlations of the four distress measure items. Of the six reflective WASH-GEM measures tested, four measures had the required divergence from *Distress* with correlation coefficients less than −0.15, $p < 0.05$ (*Household Influence, Self-efficacy, Collective Action* and *Equality Awareness*) indicating potential appropriateness for use in the PLS-SEM model. Further testing within the PLS-SEM model and review of notes from the observation data from the rapid pilot narrowed this down to two appropriate measures for identification (*Self-efficacy* with five items and *Equality Awareness* with seven items). Additionally, the intra-item correlation coefficients for the four distress measure items ranged from 0.28 to 0.58 ($p < 0.03$), falling within the anticipated thresholds and indicating low collinearity. A full correlation matrix of all 17 WASH-GEM themes can be found in S1 Fig.

## PLS-SEM model and distress measure validity

The PLS-SEM model was identified using the two selected reflective measures as illustrated in Fig 2. For both the reflective *Equality Awareness* and *Self-efficacy* measures all items were statistically significant. All but one item had sufficiently large loadings (ea7 loading of 0.191). However, as the validation of these two reflective measures was not the purpose of this analysis and they have been validated elsewhere previously [16], the item was not dropped. *Equality Awareness* and

**Table 3. Distress measure items across the datasets.**

| | Cambodia Baseline (N = 1496) | Nepal Baseline (N = 1558) | Nepal Midline (N = 1620) | Overall (N = 4674) |
|---|---|---|---|---|
| **Interview Ease (ob1)** | | | | |
| Very easy | 404 (27.0%) | 578 (37.1%) | 1169 (72.2%) | 2151 (46.0%) |
| Easy | 970 (64.8%) | 931 (59.8%) | 422 (26.0%) | 2323 (49.7%) |
| Difficult | 93 (6.2%) | 47 (3.0%) | 29 (1.8%) | 169 (3.6%) |
| Very difficult | 7 (0.5%) | 2 (0.1%) | 0 (0%) | 9 (0.2%) |
| Missing | 22 (1.5%) | 0 (0%) | 0 (0%) | 22 (0.5%) |
| **Interviewee Stress (ob2)** | | | | |
| None | 688 (46.0%) | 1100 (70.6%) | 1135 (70.1%) | 2923 (62.5%) |
| Not much | 727 (48.6%) | 364 (23.4%) | 433 (26.7%) | 1524 (32.6%) |
| Mild | 74 (4.9%) | 77 (4.9%) | 49 (3.0%) | 200 (4.3%) |
| Severe | 7 (0.5%) | 17 (1.1%) | 3 (0.2%) | 27 (0.6%) |
| Missing | 0 (0%) | 0 (0%) | 0 (0%) | 0 (0%) |
| **Interview Privacy (ob3)** | | | | |
| Very | 1080 (72.2%) | 1266 (81.3%) | 1196 (73.8%) | 3542 (75.8%) |
| Somewhat | 221 (14.8%) | 236 (15.1%) | 405 (25.0%) | 862 (18.4%) |
| Not very | 170 (11.4%) | 40 (2.6%) | 18 (1.1%) | 228 (4.9%) |
| Not at all | 23 (1.5%) | 16 (1.0%) | 1 (0.1%) | 40 (0.9%) |
| Missing | 2 (0.1%) | 0 (0%) | 0 (0%) | 2 (0.0%) |
| **Interviewee Comprehension (ob4)** | | | | |
| Very much | 296 (19.8%) | 742 (47.6%) | 1165 (71.9%) | 2203 (47.1%) |
| Somewhat | 1062 (71.0%) | 648 (41.6%) | 443 (27.3%) | 2153 (46.1%) |
| Not much | 126 (8.4%) | 125 (8.0%) | 11 (0.7%) | 262 (5.6%) |
| Not at all | 12 (0.8%) | 43 (2.8%) | 1 (0.1%) | 56 (1.2%) |
| Missing | 0 (0%) | 0 (0%) | 0 (0%) | 0 (0%) |

*Self-efficacy* had Cronbach's alpha coefficients of 0.90 and 0.88 respectively, indicating that these measures are internally consistent and applicable for identifying the model used to test the reliability of the *Distress-4* measure.

For the formative *Distress-4* measure, all items had VIF scores under the threshold of 3 (ranging from 1.180 to 1.771), indicating low collinearity as also illustrated in the intra-item correlations. Three of the four distress items had statistically significant weighting as indicated in Fig 2 and the same three items had loadings greater than 0.5 (0.79 to 0.85 p < 0.01). The exception was ob3_privacy, which had a loading of 0.40 (p < 0.01). These results recommend reflection on the importance of the item for the measure's conceptualization, however as the privacy item is important for our conceptualization of distress, we have chosen to keep the item within the measure. These same broad results also held for a non-bootstrapped model.

The path coefficients between *Equality Awareness* and *Distress* ($\beta = -0.198$, p < 0.001) and *Self-efficacy* and *Distress* ($\beta = -0.396$, p < 0.001), both performed as anticipated indicating weak negative influence from the two reflective constructs and distress. As anticipated, the overall model indicates a weak influence of the reflective measures on distress for our dataset ($r^2 = 0.203$). Using these two divergent constructs enabled us to identify a model and test the validity of the formative construct; together these results indicate that the distress measure is valid for use as a formative construct. Additionally, a reversed PLS-SEM model was developed (S2 Fig) with *Equality Awareness* and *Self-efficacy* as outcome variables ($\beta$ ranged $-0.405$ to $-0.217$, p < 0.001), with the same broad results for the *Distress-4* construct.

**Fig 1. Intra measure and item correlation matrix indicating correlation coefficients.** Explored WASH-GEM Measures: Household Influence, Household Autonomy, Self-efficacy, Collective Action, Equality Awareness and Mobility. Matrix also includes a simple sum score of the distress measure and the four distress measure items (ease, stress, privacy and comprehension).

## Scoring

The most appropriate scoring approach was identified by correlating two versions of the *Distress-4* construct scores: 1) calculated through PLS-SEM (regressions weighing) and 2) using a simple sum scoring approach. The scores correlated at r = 0.96, p > 0.001 (r = 0.93; 0.90–0.99 when disaggregating by gender and year/country), indicating that a simple score approach is justifiable. As this is much easier for project teams to calculate, we continue the analysis with the simple score approach.

## Construct validity through known groups

Construct validity was explored both at the measure (Fig 3) and item (Fig 4) levels through known groups analysis through the pooled data; the groups were chosen based on the observations from the rapid pilot. Notably at the measure level,

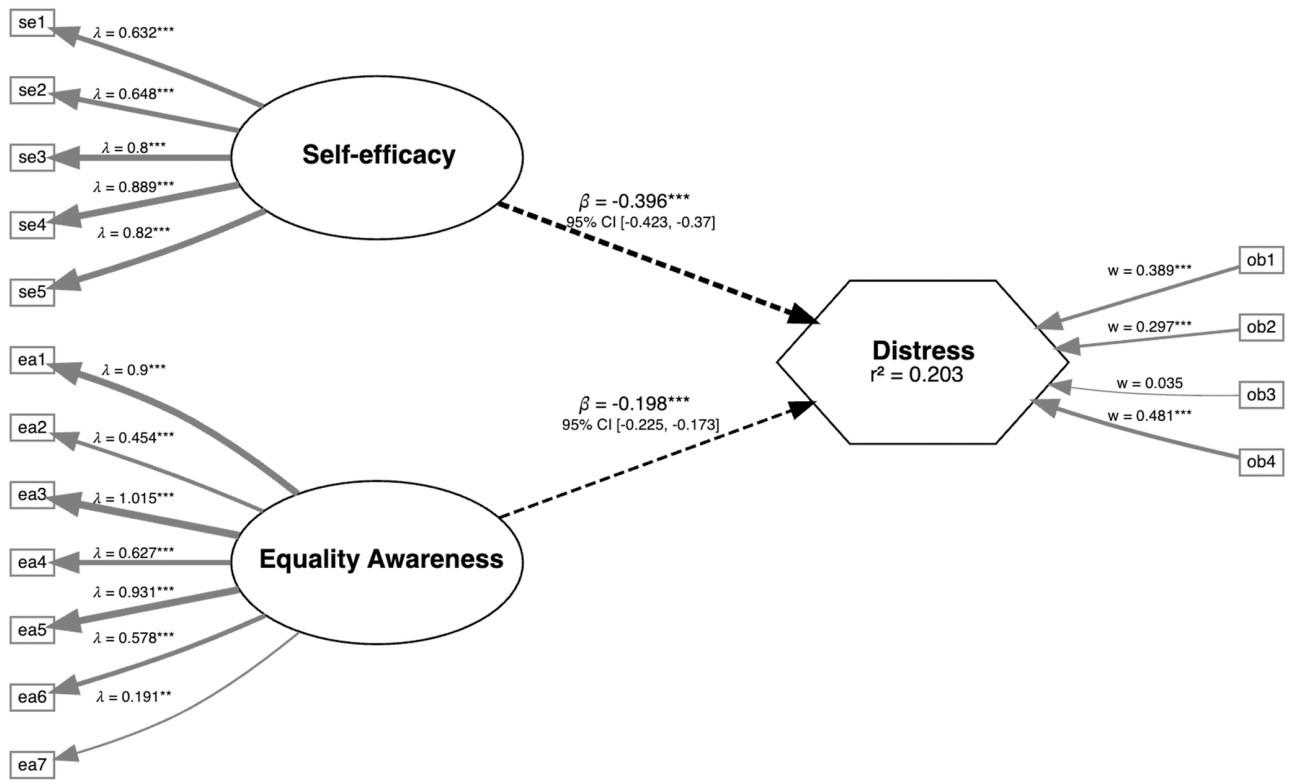

**Fig 2. PLS-SEM model illustration indicating measure weights/loadings, path coefficients, and the model's coefficient of determination.**

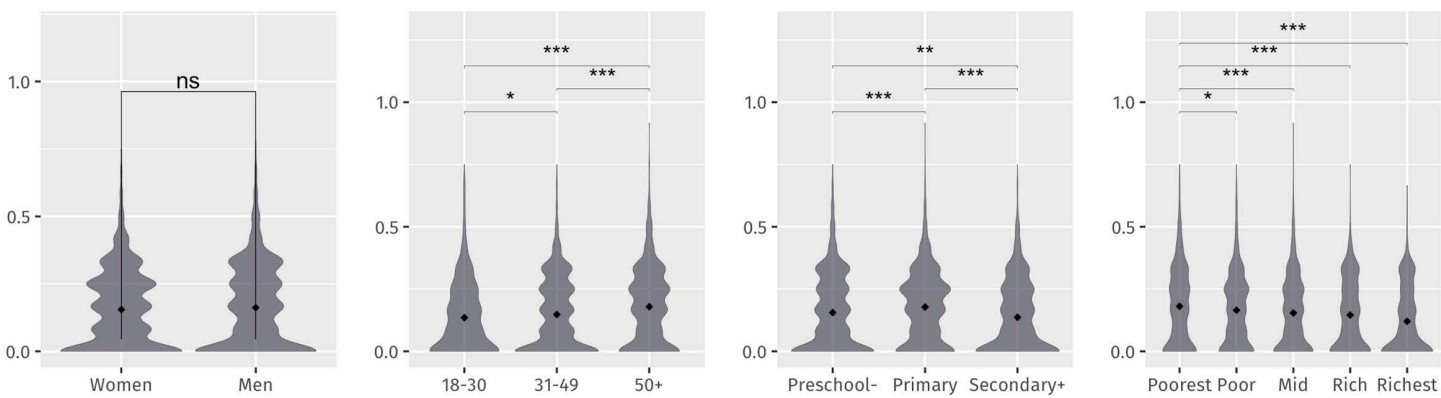

**Fig 3. Violin plots of normalized distress measure scores by gender, age, education level and wealth quintile indicating statistical significance.**

older (p < 0.001), poorer (p < 0.001), and less educated (p < 0.01), people all had statistically significant lower scores (more distress) than their younger, educated, and wealthier counterparts as illustrated in the violin plots of Fig 3. However, there was not a statistical difference between genders – an anticipated response based on other WASH-GEM studies and the rapid pilot observations.

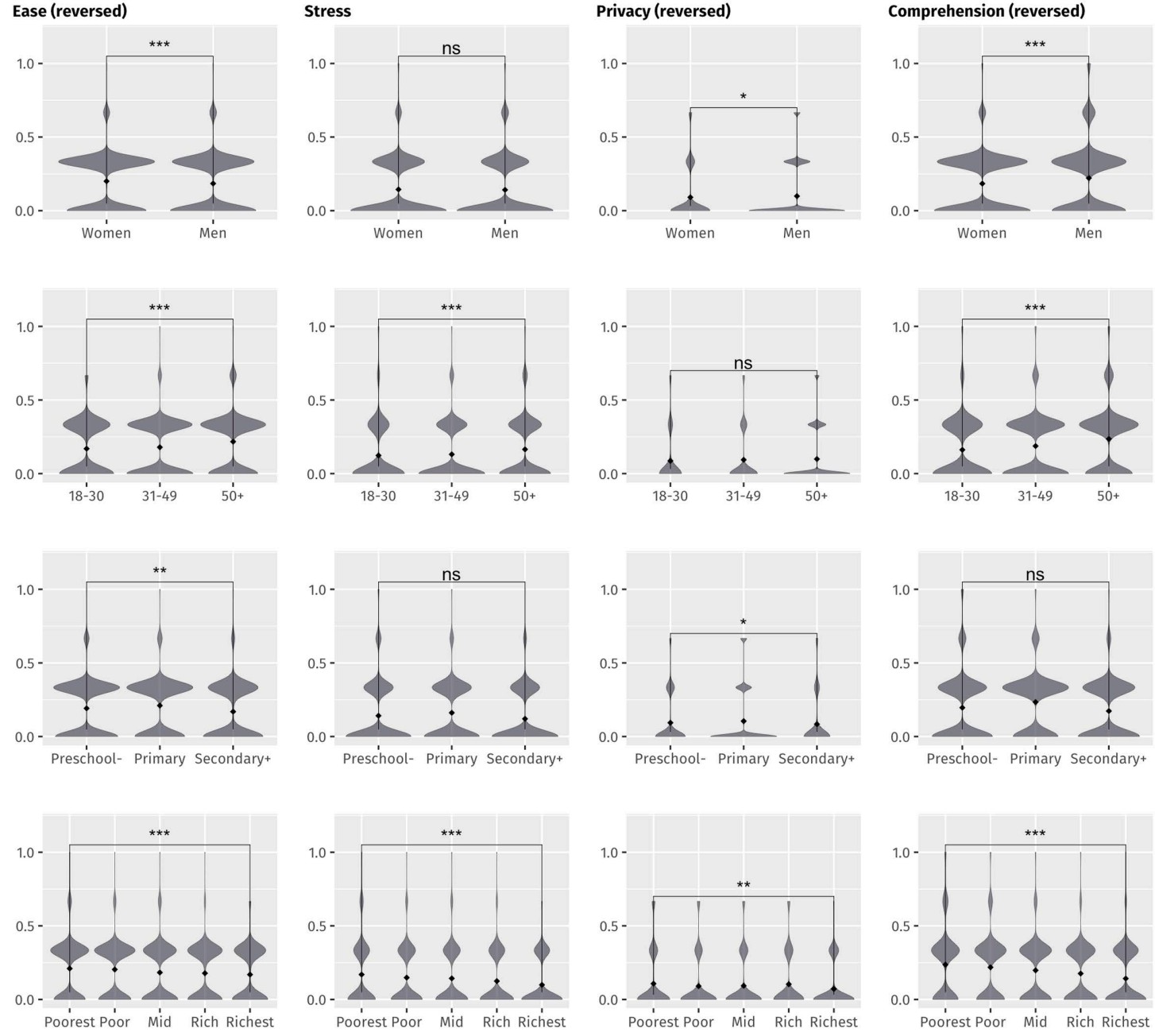

**Fig 4. Violin plots of normalized item scores by gender, age, education level and wealth quintile indicating statistical significance.**

At the item level, there were statistically significant differences across all known-groups (gender, age, education, and wealth) for ease, and across some know-groups for stress (age, wealth), privacy (gender, education, wealth), and comprehension (gender, age, wealth). Notably, there were statistically significant differences for all four items across wealth quintiles.

## Discussion

### Summary and interpretation

This study aimed to examine the validity of a rapidly developed distress measure for use in enumerator structured interview surveys – a key tenent of international development programming and research. The measure was developed in the context of fieldwork and implementation of a wider study. A PLS-SEM model was identified using two reflective constructs to explore the *Distress-4* measure's validity. The four items were shown to form a valid measure as assessed through formative measure validation procedures [26,28]: content and face validity; divergence validity; assessment of collinearity; structural validity of the measure's internal model; and construct validity through known groups testing. Overall, the measure provides a validated snapshot of four important do-no-harm assumptions related to ease, stress, privacy, and comprehension.

Additionally, the PLS-SEM model illustrates the weak but significant influence of *Equality Awareness* and *Self-efficacy* on *Distress* within the case study dataset. This highlights that within our dataset lower gender equality scores led to increased distress. We identify three potential reasons for these connections. As the survey explored aspects of equality in detail through multiple questions, it is possible that this repetition of potentially uncomfortable gender equality focused questions is associated with increased distress. It is also possible, that respondents who showed lower equality awareness picked up on subtle cues from enumerators that their responses were not aligned with a 'positive' direction. Lastly, lower equality awareness participants were more likely to be confused with phrasing and words related to gender equality as they may have been less exposed to such language. As such, we speculate that 'lower performance' on any survey tool could also theoretically lead to increased distress.

The distress measure is, to the best of our knowledge, the only tool developed to measure distress in contexts outside of clinical practice and outside of the United States and Australia, with our testing taking place in two Asian countries. As many international development surveys focus on potentially sensitive topics, the sector has an obligation to track, mitigate and reflect on the potential harm of research. Some scholars even take this further to call for research which actively aims to 'do-more-good' in a transformative approach to research and evaluation [29]. We now reflect on three use cases for the distress measure.

### During deployment

During a survey deployment the distress measure can be used to track distress by the enumeration team. Distress data can be collated on-the-spot and used as a part of the daily debriefing to identify cases of distress that may require further referral to supporting services. Such referral approaches are best practice in research on sensitive topics [22]. There are also opportunities during these debriefs to revise wording or re-order questions to mitigate stress and concerns. However, it is important to create a safe debriefing space to ensure that enumerators feel comfortable and empowered to accurately report distress.

### After deployment

After a survey deployment, within an after-action-review [30], distress measure results can be used to analyze which types of individuals were more likely to experience distress. This type of analysis can explore intersectional aspects such as age, ethnicity, education level and poverty. At this level, analysts can also explore if certain enumeration teams were more likely to report distress. Lessons from the after-action-review should be shared with enumeration teams as well as research and evaluation team members to identify ways to reduce harm in future surveys.

### Within reporting

Lastly, we argue that studies on sensitive topics should report distress results as part of their wider do-no-harm strategies for transparency and reflection. Within the positivist and post-positivist paradigms that dominant much research on

sensitive topics [31] research is seen as a neutral activity, in which enumerators are there solely to collect information from communities. However, there is increasing evidence that the very act of asking questions can raise the critical consciousness of participants and lead to both positive and negative outcomes [32]. Nevertheless, many researchers on sensitive topics are not cognizant of the potential harm of their research and as such do not reflect on the potential negative (and positive) implications. A paradigm shift towards a transformative perspective of research and evaluation, would support stronger do-no-harm approaches and enable researchers to purposely pursue positive impacts. This is particularly important in the context of research conducted in international development contexts, but also relevant in other settings.

### Limitations and future research

The validation and value of this measure must be understood alongside its limitations relating to its rapid development and novelty.

First, although the items were developed and interrogated with a group of expert practitioners and academics, there was not time or opportunity to identify a wide range of items and use other measure development procedures. As such, the four items tested in this article, were the only four tested. Nonetheless, the four items still pass all the tested validation procedures.

Second, as the measure was developed as an enumerator observation, rather than a question-based approach, there is a risk of under-reporting by enumerators. Care must be taken during enumeration training to accurately explain the purpose of distress tracking and a random sample of interviews should ideally be observed to encourage accurate reporting.

Third, as the measure was developed rapidly, there was not opportunity to date to test criterion validity by comparing other similar reflective measures which are directly asked to respondents such as the interview distress assessment [10] or the reactions to research measure [12]. Additionally, validity tools such as test-retest validity would not be appropriate for a measure seeking to understand the experiences in conducting a survey.

Future work should continue to test and refine this measure for use in wider contexts and across different types of surveys on sensitive topics.

### Conclusions

Within this article, we have presented the development and validation of a distress measure for the international development sector and research and evaluation more broadly. The four-item construct measures participant distress in survey deployment related to sensitive topics. Research and evaluation remain a critical component of development interventions and as such it is important that these activities do not cause harm to participants. Ultimately, the *Distress-4* provides a simple approach to track, mitigate and reflect on the potential harm of research, with an aim of creating research which does more good.

### Supporting information

**S1 Data. Deidentified Cleaned Dataset.** Compiled dataset for the three rounds of data collection with scored measures for WASH-GEM themes and selected items. Response options in English translation have been harmonized across three rounds of data collection.
(CSV)

**S1 Analysis Code. R-Studio Analysis Code.** Simplified code used for the distress measure validation analysis.
(RMD)

**S1 Fig. Correlation matrix of all 17 WASH-GEM themes and the four distress items.**
(TIF)

**S2 Fig. PLS-SEM model with reflective measures as outcome variables.**
(TIF)

**S1 Table. Similar distress measures.** A selection of existing measures to explore distress or similar aspects for research participants.
(DOCX)

**S1 Checklist. Inclusivity in global research checklist.**
(DOCX)

## Acknowledgments

We are grateful for all the enumerators who participated in this study and helped to clarify the importance of tracking distress. We are also grateful to all participants for their insights. Additionally, thanks to the editor and reviewers of this manuscript whose feedback have helped shape the direction of our analysis.

## Author contributions

**Conceptualization:** Jess MacArthur, Juliet Willetts.

**Formal analysis:** Jess MacArthur.

**Funding acquisition:** Juliet Willetts.

**Investigation:** Ratan Budhathoki, Min Prasad Basnet, Ambika Yadav, Sabitra Dhakal.

**Methodology:** Jess MacArthur.

**Project administration:** Ratan Budhathoki, Juliet Willetts.

**Supervision:** Ratan Budhathoki, Min Prasad Basnet, Juliet Willetts.

**Validation:** Ratan Budhathoki, Min Prasad Basnet, Ambika Yadav, Sabitra Dhakal.

**Visualization:** Jess MacArthur.

**Writing – original draft:** Jess MacArthur.

**Writing – review & editing:** Jess MacArthur, Ratan Budhathoki, Juliet Willetts.

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
