## [Decision Letter · Decision Letter 0]

28 Nov 2024

Dear Dr. MacArthur,

Thank you for submitting your manuscript to PLOS ONE. After careful consideration, we feel that it has merit but does not fully meet PLOS ONE’s publication criteria as it currently stands. Therefore, we invite you to submit a revised version of the manuscript that addresses the points raised during the review process.

**The further development of the Distress Scale is a topic of interest to researchers and clinicians. As you see below, three reviewers (including me) provided comments on the report. Reviewers provided constructive points regarding the further development of the manuscript as it stands. I highlight two things: the need to improve methodology underpinning the development of the scale, and the need to increase clarity of reporting of the current development.**

**Based on my comments and those of the reviewers, I invite authors to revise their manuscript and answer our questions.**

**My comments:**

**I miss the results and conclusion of the scale development and measurement properties in the abstract.**

**Authors state “Ultimately, these considerations aim to mitigate potential distress and reduce the extractive nature of enumerated surveys” in the introduction section. How do the considerations above reduce the extractive nature of enumerated surveys?**

**Authors state “Sample sizes are often larger than qualitative studies in order to ensure statistically significant analysis”. Rather than pointing to a significant analysis, adequate sample sizes are necessary to ensure statistical power for our study. Then our results can be generalized. Statistical significance is not directly determined by sample size alone. Please change "statistically significant analysis" to "statistical power and generalizability of results".**

**The third paragraph is unclear. Please explain how the sensitive topics mentioned can lead to distress reactions. The following statement, "However, there are limited tools to track and evaluate the comfort and ease - described in this study as distress - of participants in the enumerated surveys" is also unclear and needs to be referenced. What tools are available to assess the construct mentioned? Is the construct "comfort and ease" (of what?) or "distress" (of what?)? What type of participants (e.g., general population, adults, patients)? In the Background section, you can summarize (in 1 to 3 sentences) Table 1 in the text.**

**Please contextualize more quantitative studies. You only mention this term in your last paragraph, but the term is important throughout the manuscript.**

**Please consider presenting the Introduction and Background sections together.**

**There are some scales as described by the authors. Why should a new scale be developed?**

**What do you mean by “field research in the Global South”?**

**The subtitle “Approach” should be changed to “Methods”.**

**Did you review the literature to gain insight into the development of the scale? Did you review the ISOPOR recommendations for instrument development?**

**The methods section is unclear. Please be more specific:**

**1 - How did the team identify the four aspects of potential distress? How was this assessment conducted?**

**2 - Were the team experts in the field? If so, how were they selected and how was their expertise assessed?**

**3 - The following statement needs to be improved: "It should be noted that the Distress Scale, in a similar fashion to the Coercion Assessment Scale". What do you mean by "fashion"? What does the Coercion Assessment Scale assess?**

**4 - The “Programmatic context” paragraph is unnecessary because you already briefly stated it before.**

**5 - Details of the sampling methods (population, selection, etc.) used in the current study should be reported.**

**6 - In Table 2, the first item seems to apply more to interviewers than to participants. The other items are more directed at the participants. Is this correct? The answer/scoring is not clear. For example, how will participants score “minimal” in ob2? Why are there different numbers (1/3, 2/3)?**

**Please delete “by the first author” in the subtitle 3.4.**

**The term “Item Validity” should be changed. This is not a measurement property (http://dx.doi.org/10.1016/j.jclinepi.2010.02.006).**

**Did you assess the content validity of the scale with experts and participants? If so, how did you conduct the evaluation?**

**In your hypothesis test, poorer, older, and less educated people would have higher levels of distress than who? How high a level? Did you base your hypothesis on the literature or was it your experience?**

**Is the Distress scale based on a reflective or formative model? If it is a formative model, structural validity and internal consistency should not be assessed. If it is based on a reflective model, you can perform factor analyses and add more details to the analysis (e.g., method estimation, rotation, etc.).**

**Please change “Internal Reliability” to “Internal Consistency”, according to the recommendations (http://dx.doi.org/10.1016/j.jclinepi.2010.02.006).**

**What "key groups" did you use to assess measurement invariance?**

**Did you have missing data? If so, how did you deal with it?**

**In the Results section, you should say "Construct validity through known groups" instead of "Content validity through known groups". Please use specific guidelines/checklists to report the Results section.**

**Did you adjust the model based on the modification indices of the factor analysis? This may improve the RMSEA.**

**Why did you use a bi-factor model for scoring if the scale is unidimensional?**

**“4.7 Limitations” should be moved to the end of the Discussion section.**

**Please revise your results and discussion based on my previous comments.**

**I hope these comments are helpful.**

Please submit your revised manuscript byJan 12 2025 11:59PM. If you will need more time than this to complete your revisions, please reply to this message or contact the journal office at plosone@plos.org . A rebuttal letter that responds to each point raised by the academic editor and reviewer(s). You should upload this letter as a separate file labeled 'Response to Reviewers'.A marked-up copy of your manuscript that highlights changes made to the original version. You should upload this as a separate file labeled 'Revised Manuscript with Track Changes'.An unmarked version of your revised paper without tracked changes. You should upload this as a separate file labeled 'Manuscript'.

We look forward to receiving your revised manuscript.

Kind regards,

Guilherme Tavares de Arruda

Academic Editor

PLOS ONE

**Journal requirements:**

This work was supported by the Australian Government’s Water for Women Fund [Grant number WRA-034 - JW]. The funders did not play any role in the study. 

5. We note that you have referenced "MacArthur J, Basnet MP, Budhathoki R, et al." and "MacArthur J, Chase RP, Gonzalez D, et al." which has currently not yet been accepted for publication. Please remove this from your References and amend this to state in the body of your manuscript: (MacArthur J, Basnet MP, Budhathoki R, et al." and "MacArthur J, Chase RP, Gonzalez D, et al." [in preparation) as detailed online in our guide for authors

http://journals.plos.org/plosone/s/submission-guidelines#loc-reference-style. 

6. Please remove your figures from within your manuscript file, leaving only the individual TIFF/EPS image files, uploaded separately. These will be automatically included in the reviewers’ PDF.

Reviewers' comments:

Reviewer's Responses to Questions

**Comments to the Author**

1. Is the manuscript technically sound, and do the data support the conclusions?

Reviewer #1: Yes

Reviewer #2: No

2. Has the statistical analysis been performed appropriately and rigorously?

Reviewer #1: No

Reviewer #2: No

3. Have the authors made all data underlying the findings in their manuscript fully available?

Reviewer #1: Yes

Reviewer #2: Yes

4. Is the manuscript presented in an intelligible fashion and written in standard English?

Reviewer #1: Yes

Reviewer #2: Yes

**Reviewer #1: ** I found that the paper is well motivated. It developed a distress scale to evaluate distress in quantitative surveys. The authors proposed four aspects of potential distress and translated each of them into an item. In general, the paper is well written. I only have one concern about methodological aspect used. As stated by the authors, each of the four items represents one aspect of distress. Accordingly, the four items are formative indicators of distress. Using factor models (EFA/CFA – models for reflective indicators) for evaluating their properties is subject to misspecification and irrelevant item properties. The authors may refer to Hair et al. (2014) for assessing formative indicators.

Hair JF, Hult GTM, Tingle C, Sarstedt M, (2014) A Primer on Partial Least Squares Structural Equation Modeling (PLS-SEM), Sage.

**Reviewer #2:**  The authors conducted a study to develop a scale measuring distress caused by enumerated surveys. This is a necessary effort as discussed in the manuscript, and I appreciate the work put into the study. However, I have several methodological and technical concerns that may adversely impact the quality of the research:

1. Item indices misinterpretation:

a. On page 7, the reported item indices are about item quality rather than item validity. These indices do not address "the extent to which the items represent a range of experiences," as stated. In addition, there are different frameworks for evaluating item quality. Did the authors use Classical Test Theory (CTT) or Item Response Theory (IRT)? If IRT was used, please specify the model and report relevant software for your analysis.

2. Construct validity:

a. The study lacks an evaluation of convergent and divergent/discriminant validity, which are standard components of construct validation. Assessing group differences in distress by SES, age, and education level alone may be insufficient. In addition, if empirical studies exist that support these group differences, please discuss and cite them.

3. Known-Groups Analysis:

a. The role of known-groups analysis in informing content validity is unclear. Please clarify how this analysis supports the claim of content validity on page 11.

4. The small number of items is concerning, and the justification that “there was not time or opportunity to identify a wide range of items and use traditional scale development procedures to drop items which did not fit in the model” is problematic.

I hope these comments are helpful.

**Do you want your identity to be public for this peer review?** For information about this choice, including consent withdrawal, please see our Privacy Policy

Reviewer #1: **Yes: ** Yusep Suparman

Reviewer #2: No

---

## [Author Response · Author response to Decision Letter 1]

18 Mar 2025

Dear Reviewers,

Thank you for your helpful comments and ideas on how to strengthen our manuscript.

Based on the recommendations of the editor and reviewers, we have revised the manuscript to present the measure as a formative rather than reflective construct. We have updated the methods to use Partial Least Squares Structural Equation Modeling (PLS-SEM) to help identify and validate the distress measure.

We have aimed to address all your comments and recommendations as outlined in the response to reviewers document and look forward to your next review.

Best regards,

---

## [Decision Letter · Decision Letter 1]

8 June 2025

Quantifying participant distress: Validity and applicability of a distress measure to evaluate harm in quantitative assessments

PONE-D-24-39729R1

Dear Dr. MacArthur,

We’re pleased to inform you that your manuscript has been judged scientifically suitable for publication and will be formally accepted for publication once it meets all outstanding technical requirements.

Kind regards,

Guilherme Tavares de Arruda

Academic Editor

PLOS ONE

Additional Editor Comments (optional):

Reviewers' comments:

Reviewer's Responses to Questions

**Comments to the Author**

Reviewer #2: (No Response)

2. Is the manuscript technically sound, and do the data support the conclusions?

Reviewer #2: Partly

3. Has the statistical analysis been performed appropriately and rigorously?

Reviewer #2: Yes

4. Have the authors made all data underlying the findings in their manuscript fully available?

Reviewer #2: Yes

5. Is the manuscript presented in an intelligible fashion and written in standard English?

Reviewer #2: Yes

Reviewer #2: The structure of the paper is much clearer now and the overall quality has improved. I have a few comments remaining:

1. Divergent validity and structural validity are subtypes of construct validity. You may consider combining these sections under a broader discussion of construct validity.

2. The paragraph below on Page 22 includes a few inaccuracies: (1) Comparing your measure with other similar reflective measures would be an evaluation of convergent validity, not criterion validity, (2) Criterion validity could be assessed by examining how well your measure predicts relevant external outcomes. I would suggest adding this if you have data on external outcome variables, (3) What is test-retest validity? Did you mean test-retest reliability?

a. “Third, as the measure was developed rapidly, there was not opportunity to date to test criterion validity by comparing other similar reflective measures which are directly asked to respondents such as the interview distress assessment [10] or the reactions to research measure [12]. Additionally, validity tools such as test-retest validity would not be appropriate for a measure seeking to understand the experiences in conducting a survey”

**Do you want your identity to be public for this peer review?** For information about this choice, including consent withdrawal, please see our Privacy Policy

Reviewer #2: No

---

## [Editor Report · Acceptance letter]

PONE-D-24-39729R1

PLOS ONE

Dear Dr. MacArthur,

I'm pleased to inform you that your manuscript has been deemed suitable for publication in PLOS ONE. Congratulations! Your manuscript is now being handed over to our production team.

Kind regards,

on behalf of

Dr. Guilherme Tavares de Arruda

Academic Editor

PLOS ONE